# On Evaluating LLM Alignment by Evaluating LLMs as Judges

**Yixin Liu**[1]    **Pengfei Liu**[2]    **Arman Cohan**[1]

[1]Yale University    [2]Shanghai Jiao Tong University
{yixin.liu, arman.cohan}@yale.edu

## Abstract

Alignment with human preferences is an important evaluation aspect of LLMs, requiring them to be helpful, honest, safe, and to precisely follow human instructions. Evaluating large language models' (LLMs) alignment typically involves directly assessing their open-ended responses, requiring human annotators or strong LLM judges. Conversely, LLMs themselves have also been extensively evaluated as judges for assessing alignment. In this work, we examine the relationship between LLMs' generation and evaluation capabilities in aligning with human preferences. To this end, we first conduct a comprehensive analysis of the generation-evaluation consistency (GE-consistency) among various LLMs, revealing a strong correlation between their generation and evaluation capabilities when evaluated by a strong LLM preference oracle. Utilizing this finding, we propose a benchmarking paradigm that measures LLM alignment with human preferences *without* directly evaluating their generated outputs, instead assessing LLMs in their role as evaluators. Our evaluation shows that our proposed benchmark, ALIGNEVAL, matches or surpasses widely used automatic LLM evaluation benchmarks, such as AlpacaEval and Arena-Hard, in capturing human preferences when ranking LLMs. Our study offers valuable insights into the connection between LLMs' generation and evaluation capabilities, and introduces a benchmark that assesses alignment without directly evaluating model outputs.[1]

## 1 Introduction

Alignment with human preferences is a key property of LLMs, requiring them to accurately follow user instructions, generate responses that meet user needs, and reflect human values [29, 4]. Evaluating LLM alignment[2] typically involves human evaluations of model outputs in response to various user queries, since it requires assessing LLMs' capabilities in various open-ended tasks. However, such large-scale and reliable human evaluations are often complex, expensive, and time-consuming [46]. To scale this process, the widely used ChatBot Arena benchmark [5] relies on crowd-sourced annotations, where each instance consists of a pairwise comparison between two model outputs for a given instruction. To reduce reliance on the expensive process of human evaluation, automatic alignment benchmarks have been proposed [45, 22, 21, 24], where LLMs as judges are used in place of human annotators, enabling faster evaluation while maintaining a reasonably high level of agreement with human preferences. Consequently, this LLMs-as-Judges paradigm has been commonly used for LLM evaluation in alignment and other open-ended tasks.

---

[1]ALIGNEVAL is available at https://github.com/yale-nlp/AlignEval.

[2]In this work, we use "LLM alignment" to refer to LLMs' general capabilities in following human instructions and providing helpful, high-quality responses, which goes beyond safety or harmlessness alignment.

39th Conference on Neural Information Processing Systems (NeurIPS 2025).

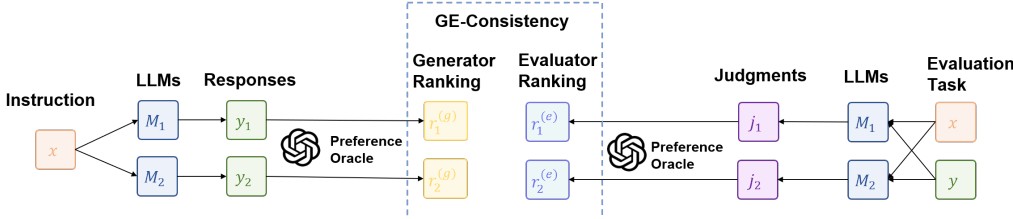

Figure 1: Illustration of Generation-Evaluation Consistency (GE-consistency), where LLMs' generation and evaluation capability rankings are compared using a preference oracle.

Understanding and evaluating how reliable LLMs are as judges has also become an important topic [43, 19, 25]. As discussed earlier, LLMs-as-Judges is an important paradigm for LLM evaluation. Furthermore, LLMs can also be used as judges in model training with preference optimization algorithms in place of fine-tuned reward models [35]. The evaluations of LLMs as judges are typically conducted by comparing the predictions of LLMs against human annotations for the same evaluation tasks. Recent studies [47, 19] have shown that frontier LLMs are strong judges for alignment evaluation. In particular, they can serve as effective out-of-the-box generative reward models [26], achieving performance competitive with fine-tuned discriminative reward models.

We argue that it is important to study the connection between these two related capabilities of LLMs: generating responses that align with human preferences and evaluating whether responses are aligned with human preferences. Understanding this connection has significant implications for both model evaluation, by revealing the (in)consistency between these capabilities [41, 23], and model training, particularly in exploring the feasibility of self-improvement where an LLM's training is supervised by its own judgments, which inherently requires a model to accurately assess its own outputs [32, 42]. Along this direction, prior work has examined the (in)consistency of individual LLMs when acting as both generator and validator [32, 42]. However, a comprehensive study is still lacking on whether the performance rankings among various LLMs are consistent between the generator and evaluator roles, i.e., whether models that rank higher in generation also tend to rank higher in evaluation.

Therefore, we conduct a comprehensive analysis of generation-evaluation consistency (GE-consistency) in LLM alignment (§3). We begin by formally defining GE-consistency (Figure 1), measured as the correlation between the performance rankings of LLMs when evaluated as generators and as evaluators (§3.1). Next, we conduct a controlled investigation to operationalize this concept using a frontier LLM (GPT-4o) as the preference oracle (gold-standard evaluator) to assess the generation and evaluation capabilities of 15 LLMs (§3.2). We then extend the analysis to additional LLMs as oracles (§3.3). The results reveal that a high degree of GE-consistency, specifically a Spearman's correlation of 0.96 between the performance rankings of LLMs as generators and as evaluators exists under specific conditions. These conditions include having a strong preference oracle (GPT-4o), using challenging evaluation instances (from Arena-Hard [21]), and applying an effective filtering strategy for selecting reliable task instances to evaluate LLMs as evaluators.

Building on this finding, we demonstrate the utility of GE-consistency by proposing a benchmarking paradigm that assesses LLM alignment with human preferences *without* directly evaluating their generated outputs (§4), as done in existing automated benchmarks like AlpacaEval [22, 9] or Arena-Hard; instead, we evaluate LLMs in their roles as evaluators. This new evaluation paradigm is therefore more cost efficient since it can make use of existing preference annotations provided by either human annotators or LLMs-as-Judges when evaluating new LLMs without requiring additional annotations. Our experiments show that the proposed benchmark, ALIGNEVAL, can match or even outperform widely used automatic evaluation benchmarks such as AlpacaEval and Arena-Hard in reflecting human preferences, using ChatBot Arena's LLM rankings as the gold standard. In particular, by combining ALIGNEVAL with IFEval [48], another evaluation benchmark that does not rely on LLM judges, our benchmark achieves a Spearman's correlation of 0.94 with ChatBot Arena rankings across 23 LLMs.

To summarize, our contribution is two-fold: (1) we present the first comprehensive analysis of generation-evaluation consistency (GE-consistency) across multiple LLMs, demonstrating a strong correlation between their capabilities as generators and evaluators under certain evaluation conditions; and (2) we propose and validate ALIGNEVAL, an effective automatic benchmarking paradigm for

assessing LLM alignment without relying on human annotators or LLM-based evaluators, achieving performance competitive with established benchmarks that utilize LLMs-as-Judges while reducing the evaluation cost. Crucially, our study emphasizes the significance of LLMs' evaluation capabilities as an essential aspect of LLM evaluation, highlighting the need for further research in this direction.

## 2 Related Work

**Evaluating LLM Alignment and Instruction-Following**. Alignment to human preferences is a key evaluation criterion for LLMs, typically assessed by examining model responses to curated instructions spanning diverse use cases [29, 4]. Such evaluations can involve expert annotators [37, 38, 7, 45] or crowd workers [8]. However, due to the high cost of human evaluation, ChatBot Arena [5], a crowd-annotated leaderboard, remains arguably the only benchmark offering human evaluations across a wide range of LLMs. Consequently, automatic evaluation benchmarks have been proposed, using LLMs as judges in place of human evaluators [45, 22, 21, 24, 18], which demonstrate a high level of correlations with human evaluations. While most evaluation methods focus on free-form outputs and thus require either human or LLM-based evaluators, alternative approaches assess instruction-following capabilities using rule-based, programmatic evaluation [48, 40, 15]. MixEval [28], in contrast, reduces reliance on LLMs-as-Judges by benchmarking models on short-answer or multiple-choice questions that are similar to user queries mined from existing benchmarks.

**Evaluating LLMs as Judges**. LLMs-as-judges is an important component for both model evaluation [22, 21] and training in terms of distillation or self-improvement [34, 42]. As a result, evaluating LLMs as judges has become an important research topic [20], with human evaluations serving as the gold standard [8, 43, 25]. In addition to instance-level studies comparing human and LLM judgments on specific instruction-output instances, system-level evaluations have also been conducted to assess whether LLMs can approximate human-derived rankings of LLMs' alignment levels [13, 11]. Moreover, LLM-judges are closely related to generative reward models (GRMs) [44, 26], and have been evaluated in reward modeling settings such as RewardBench [19]. Recent work shows that frontier LLMs, when used as judges or GRMs, are competitive with strong fine-tuned reward models [47, 10].

**Relationship Between LLMs' Generation and Evaluation Capabilities**. West et al. [41] proposes the "Generative AI Paradox" and demonstrates that LLMs can possess stronger generation capabilities than evaluation capabilities under certain circumstances. Li et al. [23], Rodriguez et al. [30] analyze generator-validator consistency (GV-consistency) and find that LLMs can behave inconsistently in these two roles. For example, an LLM may judge its own generated answer to a math problem as incorrect, or prefer a different option over its own response in a multiple-choice question-answering task. Song et al. [32] demonstrates the opposite gap, that verification is easier than generation in certain tasks, and shows that this is critical for enabling LLM self-improvement [42]. In this study, we investigate the consistency between LLMs' generation and evaluation capabilities regarding their alignment with human preferences. This measurement is related yet distinct from GV-consistency, as the evaluation task here involves assessing responses to instructions rather than validating or verifying objective correctness. We further discuss this and other differences in the next section.

## 3 Examining LLM Generation-Evaluation Consistency

Our key assumption behind evaluating LLMs' alignment by assessing their correlations with human evaluators when LLMs are acting as judges is a high consistency of their generation-evaluation capabilities – if an LLM is better at *evaluating* whether given responses align with a preference oracle (which may be human or LLM-based), its *generated* responses are expected to be better aligned with the same oracle as well. Therefore, we begin by investigating this generation-evaluation consistency.

### 3.1 Defining Generation-Evaluation Consistency

We first formally define generation-evaluation consistency in LLM alignment. Given a set of LLMs $\mathcal{M} := \{M_1, \ldots, M_N\}$, a preference oracle $J$, and a set of input instructions $\mathcal{I}$, we derive a ranking of the LLMs' generation capabilities by applying $J$ to evaluate their responses to $\mathcal{I}$. We denote this ranking as $R^{(g)} := \langle r_1^{(g)}, \ldots, r_N^{(g)} \rangle$. Here, $r_i^{(g)}$ is the rank assigned to model $M_i$, which is determined by its overall performance score aggregated across $\mathcal{I}$ assigned by the preference oracle $J$.

Similarly, a ranking of their evaluation capabilities is derived by assessing whether the LLMs produce evaluation results that match those of the preference oracle $J$ when applied to responses to the instruction set $\mathcal{I}$, typically by comparing the predictions from pointwise scoring or pairwise comparison. We denote this as $R^{(e)} := \langle r_1^{(e)}, \ldots, r_N^{(e)} \rangle$. Here, $r_i^{(e)}$ is the rank assigned to model $M_i$, which is determined by its overall prediction accuracy using the preference oracle's evaluations as the gold standard.

Then, the generation-evaluation consistency $c$ can be defined as

$$c(\mathcal{M}; \mathcal{J}, \mathcal{I}) := \mathcal{C}(R^{(g)}, R^{(e)}), \tag{1}$$

where $C$ is a certain correlation metric such as Spearman's ranking correlation coefficient. Below, we will denote Eq. 1 as GE-consistency for brevity. We note that a related characteristic of LLMs, the generator-validator consistency, or GV-consistency, has been investigated by previous studies [23, 30]. However, there is a key difference between GE-consistency and GV-consistency. Specifically, the GV-consistency is defined with a single LLM regarding its *inherent* consistency – e.g., whether an LLM acting as a validator deems its own response to a math problem correct. On the other hand, the GE-consistency is a *ranking-level* measure defined over a set of LLMs – it reflects whether an LLM that performs better at evaluation than another also performs better at generation. Therefore, although prior work has identified considerable GV-inconsistency in various LLMs [23, 30], GE-consistency remains a distinct and relatively underexplored property. A high level of GE-consistency is still possible, as it only requires a relative alignment between generation and evaluation abilities – i.e., better evaluators also tend to be better generators, even if the two capabilities are not individually consistent. Supporting this, there is evidence that more capable LLMs tend to be better judges or generative reward models [25, 47].

## 3.2 Measuring Generation-Evaluation Consistency using a Strong LLM as Preference Oracle

Having defined the GE-consistency, we conduct a case study to measure it. Specifically, we use a strong LLM as the preference oracle to judge both the generation and evaluation capabilities of a set of LLMs. This approach has two main advantages: (1) Using an LLM as the oracle (instead of humans) allows more reproducible and scalable experiments. (2) By comparing the evaluation and generation rankings, both judged by the same strong LLM, we can assess how well evaluation performance predicts generation quality. If GE-consistency is high, it suggests that LLMs' evaluation capabilities can be used to estimate their overall alignment, reducing the need for human or LLM-judge assessments on every new response. In summary, if the evaluation rankings closely match the generation rankings, as judged by the strong LLM, evaluation scores can serve as a reliable proxy for generation quality evaluated by LLMs-as-Judges.

### 3.2.1 Experimental Settings

To study the GE-consistency, we require: (1) a set of input instructions for LLMs to respond to, (2) a collection of LLMs to compare, and (3) a strong LLM to serve as the preference oracle for evaluating both generation and evaluation tasks. Using these components, we measure and compare each model's ability to generate aligned responses and to accurately evaluate other models' outputs.

**Instruction Set**. We select data sources for the instruction set (i.e., evaluation instances) $\mathcal{I}$ required to measure the GE-consistency: AlpacaEval [22, 9] and Arena-Hard [21], with 805 and 500 instructions, respectively. Both AlpacaEval and Arena-Hard are LLM evaluation benchmarks for which the instructions are carefully selected to reflect various user needs, making them suitable for our investigation.

**Evaluation Oracle**. We choose a strong frontier LLM at the time of writing, GPT-4o (gpt-4o-2024-08-06), as the evaluation oracle or judge $\mathcal{J}$, following similar practices in AlpacaEval and Arena-Hard where GPT-4[3] is chosen as the evaluator to compare LLM outputs.

**LLMs to Evaluate**. We select 15 post-trained LLMs as the LLM set $\mathcal{M}$ to be evaluated. The detailed information of these LLMs is in Appendix A. These LLMs provide succinct coverage of model sizes and families, and most are open-weight models for better reproducibility and accessibility.[4]

---

[3]gpt-4-1106-preview is the default model.

[4]Proprietary models have the risk of version updates and deprecation. Moreover, evaluating a single LLM in our study requires around 80M tokens, making the full study costs prohibitive for certain proprietary models.

**Obtaining LLM Generation Performance Ranking**. To obtain the ranking of LLMs' generation capabilities, $R^{(g)}$, we apply the evaluation oracle $\mathcal{J}$ to evaluate the LLMs' outputs for the instruction set $\mathcal{I}$. The evaluation is conducted in the manner of *pairwise comparison*, since it has been widely used in both AlpacaEval and Arena-Hard. On both AlpacaEval and Arena-Hard, we follow their settings by choosing either gpt-4-1106-preview or gpt-4-0314 as the baseline system, respectively. The different LLMs' outputs are then compared against the outputs of the baseline system, and their performance score is derived by aggregating their win rates across the instruction set. The prompt template used for the pairwise comparison is included in Appendix B, which directly requires the LLM to predict the better output given an output pair without explanations or a reasoning process. It was first introduced in Zeng et al. [43] and has demonstrated better effectiveness compared to prompts used in AlpacaEval or Arena-Hard despite its simpler format [25]. For more reliable evaluation, each output pair is evaluated twice by swapping the order of the two outputs.

**Obtaining LLM Evaluation Performance Ranking**. To derive the ranking of LLMs' evaluation capabilities, $R^{(e)}$, we propose to evaluate them using the evaluation result of the preference oracle as the ground-truth. Specifically, given an instruction $x$, a pair of outputs $y_1$ and $y_2$, and the evaluation result of the preference oracle $\mathcal{J}$, $s \in \{1, 2\}$ (the index of the better output), an LLM $M$ can then be evaluated as an evaluator by comparing its prediction $\tilde{s}$ against the ground-truth prediction $s$. Here, accuracy or inter-annotator agreement can be used as the evaluation metric. We choose to use inter-annotator agreement, specifically Cohen's Kappa, as the main metric instead of accuracy, since it can better reflect model performance when the label distribution is unbalanced.

**Instance Selection and Filtering for Evaluating LLMs' Evaluation Performance**. As discussed above, evaluating LLMs' evaluation capabilities in the pairwise comparison setting requires an instruction $x$ and a pair of outputs $(y_1, y_2)$. To construct the task instances for this evaluation, we reuse the preference oracle's annotations, i.e., the ground-truth label $s$ indicating the better system, from its comparisons between LLM outputs and those of the baseline system originally used to assess generation capabilities. Given an instruction set $\mathcal{I}$ of size $L$ and an LLM set $\mathcal{M}$ of size $N$, this yields a total of $2LN$ task instances, since each pairwise comparison is conducted twice with swapped output orderings, i.e., $y_1 \otimes y_2$ and $y_2 \otimes y_1$. We then apply a filtering process based on the self-(in)consistency of the preference oracle: if the oracle produces different predictions for $y_1 \otimes y_2$ and $y_2 \otimes y_1$, both instances $(x, y_1, y_2)$ and $(x, y_2, y_1)$ are discarded. Previous studies [36, 43, 25] have found that various LLMs exhibit a non-trivial level of such inconsistency, which we posit indicates uncertainty in the oracle's prediction. We provide further discussion in §3.2.2.

### 3.2.2 Result Analysis

Figure 2 shows the correlation of the LLMs' generation and evaluation performance on both AlpacaEval and Arena-Hard. The generation performance is measured by their win rates against the baseline system (GPT-4) as evaluated by the preference oracle, while the evaluation performance is measured by their agreement (Cohen's Kappa) with the preference oracle on the evaluation task instances after filtering as described above.

The results in Figure 2 demonstrate a 0.839 and 0.971 Spearman's rank correlation coefficient on AlpacaEval and Arena-Hard, respectively. **It indicates a relatively high level of GE-consistency among the evaluated LLMs with GPT-4o as the preference oracle, especially on Arena-Hard.**[5] The difference between AlpacaEval and Arena-Hard is likely due to the differences in instruction types in the two datasets – AlpacaEval contains more open-ended instructions, whereas Arena-Hard focuses on more challenging, technical instructions [22, 21, 24]. This makes evaluation on the latter more objective and stable.

In Appendix C, we provide further results on WildBench [24] with the same evaluation setting. Compared to Arena-Hard and AlpacaEval, WildBench offers a more balanced distribution of instruction types. On WildBench, we observe a GE-consistency of 0.938 Spearman's correlation, suggesting that **GE-consistency is a general pattern that holds across various types of instructions, including open-domain tasks.**

**Importance of Consistency Filtering**. As discussed in §3.2.1, the task instances used to evaluate LLMs' evaluation performance are filtered based on whether the preference oracle produces consistent predictions when the order of outputs in the pairwise comparison is swapped. On AlpacaEval

---

[5]Appendix D shows the stability of this consistency.

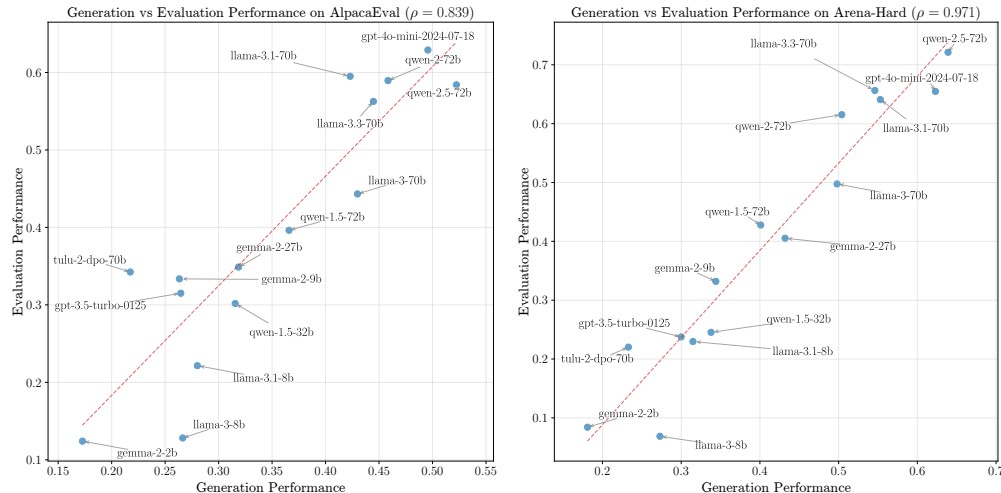

Figure 2: Generation and evaluation performance of various LLMs with gpt-4o-2024-08-26 as the preference oracle. The X-axis shows the generation performance in terms of LLMs' win rates against the baseline system (GPT-4) evaluated by the preference oracle. The Y-axis shows the evaluation performance in terms of LLMs' agreement rate (Cohen's Kappa) with the preference oracle on filtered evaluation task instances.

and Arena-Hard, 58.3% and 50.7% of instances, respectively, are filtered out due to inconsistent predictions from the oracle. **Table 1 highlights the importance of this filtering step, as the correlation between generation and evaluation rankings drops significantly without it.** This is likely because the filtering process removes two types of unreliable instances: (1) cases where the outputs are too similar to allow for an objective preference, and (2) cases where the preference oracle is uncertain and effectively guessing.

**Remark**. The high level of GE-consistency observed in this experiment, with Arena-Hard as the instruction set and GPT-4o as the preference oracle, suggests the possibility of replacing the LLM-as-Judge evaluation method with evaluating LLMs as judges for LLM alignment evaluation. Specifically, since an LLM's performance as an evaluator strongly correlates with its generation performance, we can construct an

Table 1: The impact of consistency filtering on the measurement of GE-consistency.

|  | **Spearman's Rank Correlation** | |
|---|---|---|
|  | **AlpacaEval** | **Arena-Hard** |
| w/o filtering | 0.743 | 0.793 |
| w/ filtering | 0.839 | 0.971 |

evaluation benchmark using a fixed set of evaluation task instances annotated by the preference oracle to assess LLM alignment. This allows evaluating various LLMs without requiring an LLM judge to directly assess their generated outputs. In §4, we conduct further investigation along this direction.

## 3.3 GE-Consistency with Various LLMs as Preference Oracle

In §3.2.2, the GE-consistency is measured with GPT-4o as the preference oracle. We now extend this analysis to other LLMs as preference oracles, especially the ones that are less capable than GPT-4o, to have a more comprehensive evaluation. To this end, we use the 15 LLMs ($\mathcal{M}$) evaluated in §3.2.1 as the preference oracles, which allows us to reuse the collected data annotations. The detailed experimental settings are in Appendix D.

Figure 3 shows the evaluation results: (1) The GE-consistency varies with the choice of preference oracle, with larger, more capable LLMs leading to higher GE-consistency in general. For example, the GE-consistency with llama-3-70b as the preference oracle on Arena-Hard is approximately 0.9 in terms of Spearman's correlation. (2) The GE-consistency on Arena-Hard is generally higher than the GE-consistency on AlpacaEval, aligning with the trend observed with GPT-4o as the preference oracle in §3.2.2.

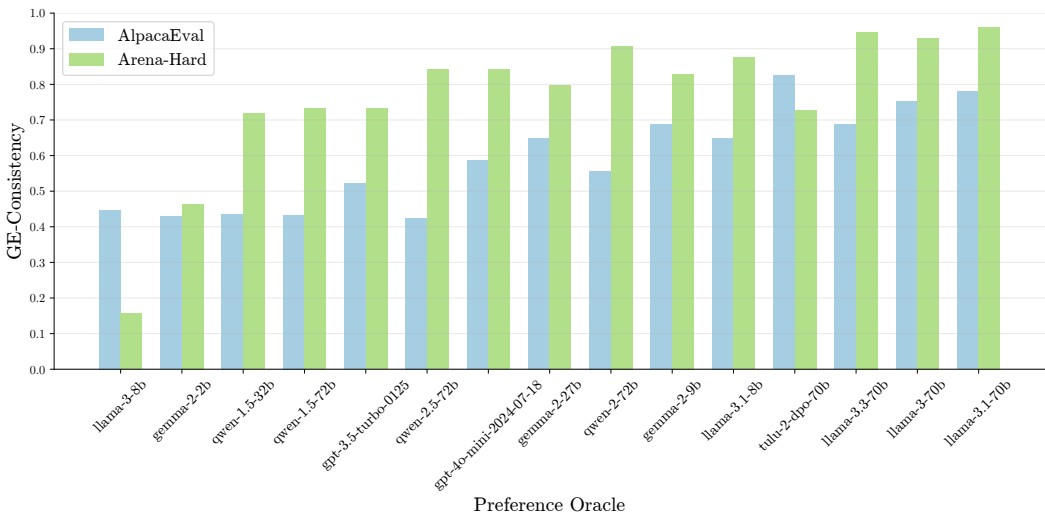

Figure 3: The GE-consistency with different LLMs as the preference oracle. Spearman's correlation between the generation and evaluation capability rankings of LLMs under different preference oracles is shown on the Y-axis. The preference oracles are sorted in ascending order of their corresponding GE-consistency levels on the X-axis.

**Remark**. The GE-consistency observed in this analysis is generally much lower than the GE-consistency with GPT-4o as the preference oracle, especially with small models such as llama-3-8b. This highlights the impact of the preference oracle on GE-consistency – intuitively, a weaker preference oracle that produces random evaluations will yield zero GE-consistency, whereas stronger LLMs as the oracle are more likely to produce stable and meaningful results.

## 4 Evaluating LLM Alignment by Evaluating LLM Evaluation Capabilities

Our analysis in §3 shows that there exists a high level of consistency among LLMs' generation and evaluation capabilities, given a strong LLM as the preference oracle. Therefore, we now investigate whether an evaluation benchmark that measures LLMs' evaluation capabilities can be used to evaluate the LLMs' alignment to *human* preferences. In other words, such a benchmark aims to achieve the same goal as the automatic LLM benchmarks such as AlpacaEval and Arena-Hard, without using LLMs as judges to evaluate LLMs' outputs. To this end, we first introduce the designed benchmark, ALIGNEVAL, in §4.1, and the experimental settings in §4.2, then provide an in-depth analysis in §4.3.

### 4.1 Building ALIGNEVAL

§3.2.2 has demonstrated that a high GE-consistency among various LLMs when GPT-4o is used as the preference oracle and Arena-Hard is selected as the instruction set. We can then construct an evaluation benchmark for assessing LLMs' evaluation capabilities, where each instance consists of an instruction, a pair of outputs, and a label indicating the preference oracle's judgment of the better output. For the input set, we reuse the instances from §3.2.1, consisting of pairwise comparisons between 15 LLMs and the baseline system on Arena-Hard, filtered to retain only those where GPT-4o produces consistent results. To further reduce computational cost, only one task instance is kept for each pairwise comparison by randomly selecting a single output order. This results in 2671 input instances in total. Regarding the gold-standard labels, apart from the annotations of GPT-4o, we also obtain the evaluation results from Claude-3.7-Sonnet [2], a recently released frontier LLM. We name the two versions of ALIGNEVAL as ALIGNEVAL-GPT and ALIGNEVAL-CLAUDE accordingly.

A key advantage of ALIGNEVAL over benchmarks like AlpacaEval or Arena-Hard is that it does not require an LLM judge to evaluate new models. Once constructed using a preference oracle, ALIGNEVAL instances can be reused to benchmark any LLM's evaluation performance, significantly reducing cost – especially compared to setups like Arena-Hard that involve expensive API calls to

proprietary LLMs. This evaluation paradigm can also extend to human preferences, where annotations are collected once and reused for evaluating future models. We compare different automatic alignment benchmarks in Table 2 and further discuss the experimental settings in §4.2.

## 4.2 Experimental Settings

**LLMs to Benchmark**. To enable a more robust and comprehensive evaluation, apart from the 15 LLMs evaluated in §3.2.1, 8 additional LLMs are added to the set of LLMs to be benchmarked. These include strong models such as Gemini-2.0-Flash[6] and Llama-3.1-405B-Instruct [14]. The selected LLMs span a range of model families, sizes, and capabilities to ensure diversity and representativeness. The detailed information of these LLMs is in Appendix A.

**Evaluation of Automatic Benchmarks**. To evaluate and compare the effectiveness of automatic benchmarks in measuring LLMs' alignment to human preferences, we use the human evaluation results in ChatBot Arena Leaderboard[7] [5] as the gold standard ranking of LLMs' alignment level. The automatic benchmarks can then be evaluated by measuring the correlation of their produced LLM rankings with the gold-standard ranking. We note that Chatbot Arena is not a true "gold standard," given its opaque data collection process and potential biases [31]. Nonetheless, we adopt it due to the lack of better alternatives and to remain consistent with common practice [22, 21, 24, 28]. Moreover, to mitigate possible biases, we use its *style-controlled* rankings.[8]

Below, we introduce the baseline benchmarks compared, with Table 2 summarizing their differences.

**Baselines: Benchmarks using LLMs as Judges**. Both **AlpacaEval** and **Arena-Hard** under their default configurations are included as baselines. For AlpacaEval, both the raw win rate and the length-controlled win rate are reported. For Arena-Hard, both the raw scores and the style-controlled scores are reported. In addition, we compare evaluation results using **GPT-4o-as-Judge** on Arena-Hard, following the same setup as in §3.2.1. This serves as the upper bound for ALIGNEVAL-GPT, which aims to approximate these evaluation results.

Table 2: Comparison of alignment evaluation benchmarks in terms of the number of instances, the need for an LLM-judge, and the estimated cost of proprietary LLM APIs for evaluating one LLM.

| Benchmark | #Instances | LLM-judge | API Cost |
|---|---|---|---|
| AlpacaEval | 805 | ✓ | $10 |
| Arena-Hard | 500 | ✓ | $20 |
| GPT4o-Judge | 500 | ✓ | $2 |
| IFEval | 541 | ✗ | $0 |
| MixEval | 1000 | ⅄ | $0.1 |
| HelpSteer3 | 4188 | ✗ | $0 |
| ALIGNEVAL | 2671 | ✗ | $0 |

**Baselines: Ground-Truth-Based Benchmarks**. Instead of relying on the LLMs-as-Judges evaluation paradigms, benchmarks can also be constructed based on (verifiable) ground-truth. Among them, **IFEval** [48] evaluates LLMs on their ability to follow specific instructions, using rule-based and programmatic evaluation methods, which does not require an LLM as a judge. It has both *loose* and *strict* grading schemas. **MixEval** [28] constructs task instances by matching real-world user queries with test examples from existing benchmarks. Evaluation compares model responses against gold-standard answers. The task instances include multiple-choice questions, which do not require an LLM judge, and short-answer questions, where an LLM judge is required, but only to compare outputs, making the task simpler and less computationally intensive than in settings like Arena-Hard. The MixEval-Hard subset is selected due to its better effectiveness.

**Baseline: LLM-Judge Evaluation using Human Preferences as Gold-Standard**. ALIGNEVAL evaluated LLMs' evaluation capabilities using strong LLMs as the preference oracle. A similar evaluation method is to use human preferences as the gold standard. To this end, we use the human annotations in **HelpSteer3**[9][39] to construct a human-preference dataset for evaluating LLMs as judges. Specifically, each instance in HelpSteer3 contains pairwise comparison annotations of two outputs for a given instruction, collected from multiple human annotators. To reduce annotation noise, we retain only instances with a preference strength of 3, indicating that one output is clearly preferred over the other based on their data collection protocol.

---

[6]https://cloud.google.com/vertex-ai/generative-ai/docs/models/gemini/2-0-flash

[7]https://lmarena.ai/?leaderboard

[8]https://lmsys.org/blog/2024-08-28-style-control/. The ranking snapshot was taken on April 18, 2025.

[9]https://huggingface.co/datasets/nvidia/HelpSteer3

Table 3: LLM Performance on Various Benchmarks. The LLMs are ordered by their style-controlled ChatBot Arena Rankings. For AlpacaEval, both the raw and the length-controlled (LC) scores are reported. For Arena-Hard, both the raw and the style-controlled (SC) scores are reported. The highest score for each benchmark is shown in **bold**. The top-3 scores are underlined.

| Model | ChatBot-Arena | AlpacaEval Raw | LC | Arena-Hard Raw | SC | GPT4o-Judge | MixEval | IFEval | HelpSteer3 | ALIGNEVAL GPT | CLAUDE |
|---|---|---|---|---|---|---|---|---|---|---|---|
| gemini-2.0-flash | 10 | **71.7** | 53.9 | **83.0** | 73.3 | 66.7 | 63.3 | 91.5 | 69.5 | **80.8** | 73.4 |
| gemini-2.0-flash-lite | 15 | 66.0 | 48.6 | 80.3 | 69.0 | 64.6 | 51.7 | 90.7 | 65.5 | 72.9 | 70.1 |
| gpt-4o-2024-05-13 | 17 | 45.5 | 55.4 | 76.0 | 73.3 | **68.7** | 62.2 | 87.8 | 71.3 | 80.4 | 67.0 |
| claude-3.5-sonnet | 18 | 32.0 | 46.5 | 75.5 | **83.2** | 67.0 | 65.5 | 87.2 | 67.7 | 70.0 | **73.9** |
| llama-3.1-405b | 22 | 33.6 | 39.2 | 65.7 | 67.2 | 57.5 | **66.0** | 89.3 | 73.1 | 74.0 | 68.2 |
| llama-3.3-70b | 37 | 39.7 | 38.6 | 62.9 | 61.0 | 54.6 | 60.9 | 92.2 | 70.1 | 74.1 | 67.2 |
| gpt-4o-mini-2024-07-18 | 37 | 38.1 | 49.2 | 74.5 | 70.4 | 62.3 | 47.6 | 84.3 | 66.1 | 74.5 | 67.0 |
| claude-3.5-haiku | 38 | 27.7 | 46.4 | 72.1 | 81.8 | 59.2 | 54.8 | 83.8 | 58.0 | 62.0 | 62.0 |
| gemini-1.5-flash | 41 | 52.3 | **57.0** | 80.2 | 77.5 | 59.6 | 47.4 | 89.3 | 60.2 | 67.6 | 61.4 |
| qwen-2.5-72b | 51 | 50.0 | 47.7 | 77.4 | 69.0 | 63.9 | 52.6 | 87.3 | 70.7 | 79.2 | 72.7 |
| llama-3.1-70b | 56 | 32.1 | 35.6 | 56.7 | 56.5 | 55.3 | 62.2 | 87.9 | 69.3 | 71.7 | 64.8 |
| gemma-2-27b | 58 | 30.3 | 47.4 | 51.5 | 49.1 | 43.2 | 47.0 | 81.7 | 49.1 | 47.9 | 43.6 |
| llama-3-70b | 65 | 36.0 | 38.1 | 51.9 | 55.5 | 49.8 | 57.0 | 84.1 | 58.8 | 55.1 | 50.2 |
| mistral-small-24b | 71 | 45.6 | 48.6 | 70.3 | 63.5 | 59.6 | 49.5 | 80.0 | 63.9 | 73.8 | 66.4 |
| qwen-2-72b | 73 | 43.0 | 48.7 | 59.2 | 59.8 | 50.4 | 53.3 | 83.5 | 67.9 | 69.8 | 66.0 |
| gemma-2-9b | 74 | 29.1 | 44.9 | 40.5 | 38.6 | 34.4 | 38.5 | 75.4 | 49.3 | 38.6 | 35.8 |
| qwen-1.5-72b | 92 | 24.2 | 34.3 | 36.3 | 44.6 | 40.1 | 45.7 | 61.8 | 46.1 | 51.6 | 47.3 |
| llama-3-8b | 96 | 18.6 | 20.0 | 21.3 | 25.8 | 27.3 | 36.7 | 77.1 | 23.5 | 6.4 | 4.9 |
| gpt-3.5-turbo-0125 | 100 | 10.8 | 22.6 | 26.3 | 43.5 | 30.0 | 41.4 | 72.5 | 37.2 | 27.0 | 23.3 |
| llama-3.1-8b | 103 | 23.9 | 24.4 | 28.7 | 27.7 | 31.5 | 44.8 | 79.6 | 36.5 | 25.8 | 24.4 |
| qwen-1.5-32b | 104 | 22.9 | 27.8 | 27.5 | 35.6 | 33.8 | 39.6 | 56.5 | 43.1 | 26.2 | 24.8 |
| gemma-2-2b | 110 | 27.1 | 30.0 | 18.9 | 15.5 | 18.1 | 26.2 | 60.8 | 16.1 | 10.7 | 10.5 |
| tulu-2-dpo-70b | 119 | 17.1 | 22.1 | 16.1 | 22.0 | 23.3 | 44.7 | 61.7 | 40.7 | 25.5 | 22.8 |

**ALIGNEVAL+: Combining ALIGNEVAL with IFEval**. Both ALIGNEVAL and IFEval evaluate LLM alignment without relying on LLMs as judges, and they are complementary in nature – ALIGNEVAL assesses an LLM's understanding of what constitutes a well-aligned output, analogous to a planning step, while IFEval evaluates precise instruction-following ability, analogous to an execution step. We refer to this combined benchmarking approach as ALIGNEVAL+, which evaluates LLMs by averaging their rankings from ALIGNEVAL and IFEval.

## 4.3   Result Analysis

Table 3 presents the evaluation results of the 23 LLMs across different benchmarks, ordered by their style-controlled rankings on ChatBot Arena, offering a detailed view of how each benchmark correlates with human evaluations. We detail our main findings below:

(1) We observe that models with high ChatBot Arena rankings generally perform well across benchmarks. For example, gemini-2.0-flash, the top-ranked LLM in our study, ranks in the top 3 on 8 out of 10 benchmarks.

(2) We observe a self-preference bias in ALIGNEVAL: ALIGNEVAL-GPT ranks gpt-4o-2024-05-13 second, while ALIGNEVAL-CLAUDE ranks claude-3.5-sonnet highest. A similar bias appears in the LLMs-as-Judges evaluation with gpt-4o-2024-08-06, which ranks its earlier version first. While expected, such bias may be reduced using multiple preference oracles. Notably, both ALIGNEVAL variants consistently place gemini-2.0-flash in the top two.

Table 4 presents the performance of different automatic alignment benchmarks regarding their correlations with the style-controlled ChatBot Arena LLM ranking, with and without being used together with IFEval(-Loose). We note two important findings:

(1) IFEval, especially with the loose grading criteria (IFEval-Loose), achieves strong performance, despite being originally developed for evaluating formatting instruction-following.

(2) Notably, **ALIGNEVAL combined with IFEval performs better or comparably to LLM-judge-based benchmarks**. This demonstrates that ALIGNEVAL+ is an effective automatic evaluation benchmark for LLM alignment that does not rely on LLM judges.

Regarding the benchmark performance without IFEval, we observe the following:

(1) **ALIGNEVAL, and especially ALIGNEVAL-CLAUDE, achieves comparable or superior performance to LLMs-as-Judges-based benchmarks.**

(2) Compared to ground-truth-based benchmarks, **ALIGNEVAL significantly outperforms MixEval.** This suggests that assessing models' evaluation capabilities on instruction outputs, as done in ALIGNEVAL, is more effective than MixEval's approach of using similar benchmark examples.

(3) **Benchmarking LLMs as judges on Help-Steer3 is relatively less effective than on ALIGNEVAL.** One likely reason is that its instruction set is not as carefully curated as those in alignment benchmarks like Arena-Hard. Prior work has proposed methods for curating high-quality instruction sets [21, 24, 28], which could improve HelpSteer3 through instance filtering, a direction we leave for future work.

Table 4: Spearman's correlation of LLM benchmarks with ChatBot Arena rankings, with and without rank averaging with IFEval-Loose. Arena-Hard-SC is the style-controlled score, AlpacaEval-LC is the length-controlled score.

| Benchmark | w/ IFEval | w/o IFEval |
|---|---|---|
| IFEval(-Loose) | 0.919 | 0.919 |
| IFEval-Strict | 0.911 | 0.880 |
| Arena-Hard | 0.946 | 0.905 |
| Arena-Hard-SC | 0.936 | 0.882 |
| AlpacaEval | 0.891 | 0.761 |
| AlpacaEval-LC | 0.925 | 0.746 |
| GPT4o-Judge | 0.958 | 0.911 |
| MixEval | 0.900 | 0.816 |
| HelpSteer3 | 0.904 | 0.813 |
| ALIGNEVAL-GPT | 0.946 | 0.856 |
| ALIGNEVAL-CLAUDE | 0.946 | 0.885 |

(4) **All alignment benchmarks show lower correlations with ChatBot Arena than reported at release**, especially AlpacaEval and MixEval (Appendix E). This is likely due to the stronger LLMs evaluated here, which may make the task more challenging than in earlier studies.

## 5 Discussion and Conclusion

The results in §4 demonstrate the promising effectiveness of ALIGNEVAL for evaluating LLM alignment. However, we emphasize that it is a proxy evaluation by design and may be vulnerable to adversarial attacks. For example, fine-tuning an LLM to act as a judge could artificially boost its ALIGNEVAL ranking without meaningfully improving its alignment. Combining ALIGNEVAL with IFEval helps mitigate this risk, as it requires the model to still be able to generate accurate responses to instructions. Still, this does not fully eliminate the concern, and we leave the development of more robust evaluation settings for future work. Nonetheless, ALIGNEVAL remains a valuable benchmark for benign evaluators, such as model developers, to assess and better understand LLM alignment and capabilities without relying on human or LLM judges for iterative output annotations.

We argue that a more critical insight of our study is that **the capability of LLMs to evaluate whether outputs align with human preferences is itself an important aspect of LLM evaluation**. Specifically, in §3, we demonstrate a strong correlation between generation and evaluation capabilities of LLMs when assessed using a strong preference oracle; in §4, we leverage this finding to construct a practical LLM evaluation benchmark. Moreover, the studied property, generation-evaluation consistency, has broader implications for both training and evaluation of LLMs. For example, if this consistency holds during training, it suggests the feasibility of self-improvement: a stronger model could better supervise its own training process. We call for future work to further investigate and better understand GE-consistency and its implications for LLM development.

## Acknowledgements

We are grateful for the TPU compute support provided by the Google TRC program and for the OpenAI API credits support provided by OpenAI's Researcher Access Program.

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

# A  Detailed Information about Benchmarked LLMs

| Name | Size | License | Reference |
|------|------|---------|-----------|
| llama-3-8b | 8b | llama3 Community | |
| llama-3-70b | 70b | llama3 Community | |
| llama-3.1-8b | 8b | llama3.1 Community | Meta AI [27], Dubey et al. [7] |
| llama-3.1-70b | 70b | llama3.1 Community | |
| llama-3.3-70b | 70b | llama3.3 Community | |
| gemma-2-2b | 2b | Gemma | |
| gemma-2-9b | 9b | Gemma | Team et al. [33] |
| gemma-2-27b | 27b | Gemma | |
| tulu-2-dpo-70b | 70b | AI2 ImpACT Low-risk | Ivison et al. [16] |
| qwen-1.5-32b | 32b | Qianwen | |
| qwen-1.5-72b | 72b | Qianwen | |
| qwen-2-72b | 72b | Qianwen | Bai et al. [3] |
| qwen-2.5-72b | 72b | Qianwen | |
| gpt-3.5-turbo-0125 | - | Proprietary | Achiam et al. [1] |
| gpt-4o-mini-2024-07-18 | - | Proprietary | |

Table 5: 15 LLMs evaluated in §3.2.1.

| Name | Size | License | Reference |
|------|------|---------|-----------|
| llama-3.1-405b | 405b | llama3.3 Community | Dubey et al. [7] |
| gpt-4o-2024-05-13 | - | Proprietary | Achiam et al. [1] |
| claude-3-haiku | - | Proprietary | Claude [6] |
| claude-3.5-sonnet | - | Proprietary | |
| gemini-2.0-flash-lite | - | Proprietary | |
| gemini-2.0-flash | - | Proprietary | Gemini et al. [12] |
| gemini-1.5-flash | - | Proprietary | |
| mistral-small-24b | 24b | Apache 2.0 | Jiang et al. [17] |

Table 6: 8 additional LLMs evaluated in §4.2.

Table 5 contains the 15 LLMs evaluated in §3.2.1. Table 6 contains the 8 additional LLMs evaluated in §4.2.

# B  Prompt Template for Evaluating LLMs as Judges

The prompt template for evaluating LLMs as judges is shown in Figure 4.

# C  GE-Consistency with GPT-4o as the Preference Oracle on WildBench

In §3.2.2, we examined the GE-consistency on AlpacaEval and Arena-Hard across 15 LLMs using GPT-4o as the preference oracle. The results show that the observed GE-consistency is higher on Arena-Hard than on AlpacaEval, specifically, a 0.971 Spearman's rank correlation versus a 0.839 correlation. Here, we conduct an examination under the same setting on WildBench [24].

Figure 5 demonstrates a 0.939 Spearman's correlation on WildBench regarding the GE-consistency. Compared to Arena-Hard and AlpacaEval, WildBench offers a more balanced distribution of instruction types, for example, 14% involve creative writing, 12% involve reasoning, and 17% involve information seeking. On this more diverse instruction set, the observed correlation is still relatively strong, indicating that a high GE-consistency is a general pattern that holds across various types of instructions, including open-domain tasks.

```
Base

[System Message]
You are a helpful assistant in evaluating the quality of the outputs for a given instruction. Your goal is to
select the best output for the given instruction.

[User Message]
Select the Output (a) or Output (b) that is better for the given instruction. The two outputs are generated
by two different AI chatbots respectively.

Here are some rules of the evaluation:
(1) You should prioritize evaluating whether the output honestly/precisely/closely executes the
instruction, then consider its helpfulness, accuracy, level of detail, harmlessness, etc.
(2) Outputs should NOT contain more/less than what the instruction asks for, as such outputs do NOT
precisely execute the instruction.
(3) You should avoid any potential bias and your judgment should be as objective as possible. For
example, the order in which the outputs were presented should NOT affect your judgment, as Output (a)
and Output (b) are equally likely to be the better.

Do NOT provide any explanation for your choice.
Do NOT say both / neither are good.
You should answer using ONLY "Output (a)" or "Output (b)". Do NOT output any other words.

# Instruction:
{INSTRUCTION}

# Output (a):
{OUTPUT_1}

# Output (b):
{OUTPUT_2}

# Which is better, Output (a) or Output (b)?  Your response should be either "Output (a)" or
"Output (b)":
```

Figure 4: Prompt template for evaluating LLMs as Judges.

# D   Detailed settings for Evaluating GE-Consistency with Various LLMs as Preference Oracle

Here, we outline the detailed settings for using various LLMs as preference oracles in measuring GE-consistency (§3.3). When calculating the GE-consistency (Eq. 1) with a certain LLM $M$ as the preference oracle $J$, we remove it from the LLM set to be evaluated ($\mathcal{M}$), since it will achieve perfect evaluation performance with itself as the evaluation oracle. This process might lead to a discrepancy in evaluation settings since the resulting LLM set $\mathcal{M} \setminus \{M\}$ is different for each LLM $M$. However, we find that the GE-consistency measured with GPT-4o as the preference oracle remains stable when any specific LLM is removed from the LLM set $\mathcal{M}$, which is detailed in Appendix D. Therefore, the impact of this process on the GE-consistency should be moderate, allowing a meaningful comparison of different LLMs as a preference oracle. In Figure 6, we show that the GE-consistency measured with GPT-4o in §3.2.2 is quite stable when any of the LLMs is removed from the LLM set to be evaluated. This result ensures the reliability of the analysis in §3.3 where one LLM is removed from the LLM set.

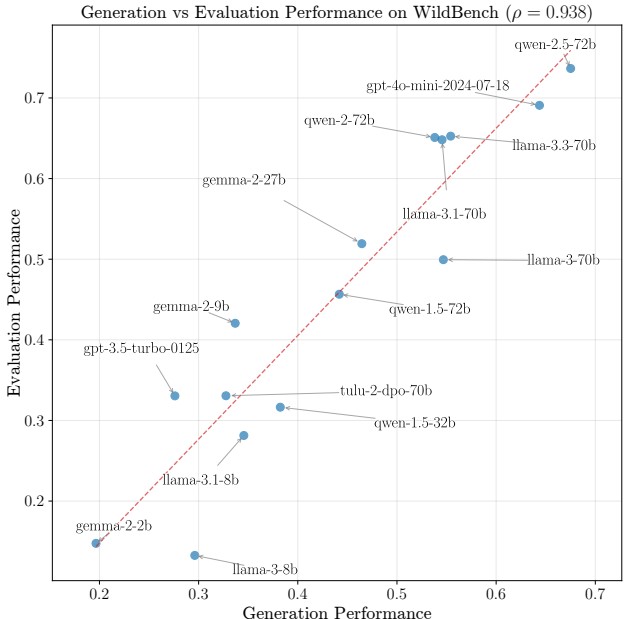

Figure 5: Generation and evaluation performance of various LLMs with gpt-4o-2024-08-26 as the preference oracle. The X-axis shows the generation performance in terms of LLMs' win rates against the baseline system (GPT-4) evaluated by the preference oracle. The Y-axis shows the evaluation performance in terms of LLMs' agreement rate (Cohen's Kappa) with the preference oracle on filtered evaluation task instances.

Table 7: Comparison of different automatic evaluation benchmarks regarding their Spearman's correlation with non-style-controlled ChatBot Arena rankings, with and without rank averaging with IFEval-Loose. Arena-Hard-SC is the style-controlled score, AlpacaEval-LC is the length-controlled score.

| Benchmark | w/ IFEval | w/o IFEval |
|---|---|---|
| IFEval(-Loose) | 0.893 | 0.893 |
| IFEval-Strict | 0.907 | 0.895 |
| Arena-Hard | 0.952 | 0.936 |
| Arena-Hard-SC | 0.914 | 0.857 |
| AlpacaEval | 0.916 | 0.839 |
| AlpacaEval-LC | 0.950 | 0.815 |
| GPT4o-Judge | 0.958 | 0.911 |
| MixEval | 0.829 | 0.727 |
| HelpSteer3 | 0.871 | 0.781 |
| ALIGNEVAL-GPT | 0.922 | 0.847 |
| ALIGNEVAL-CLAUDE | 0.916 | 0.848 |

# E    LLM Benchmark Correlations with Non-style-controlled ChatBot Arena Rankings

In §4.3, the effectiveness of various LLMs is evaluated against the *style-controlled* LLM rankings from ChatBot Arena. Table 7 instead shows their correlations with the non-style-controlled rankings, which indicates a similar trend.

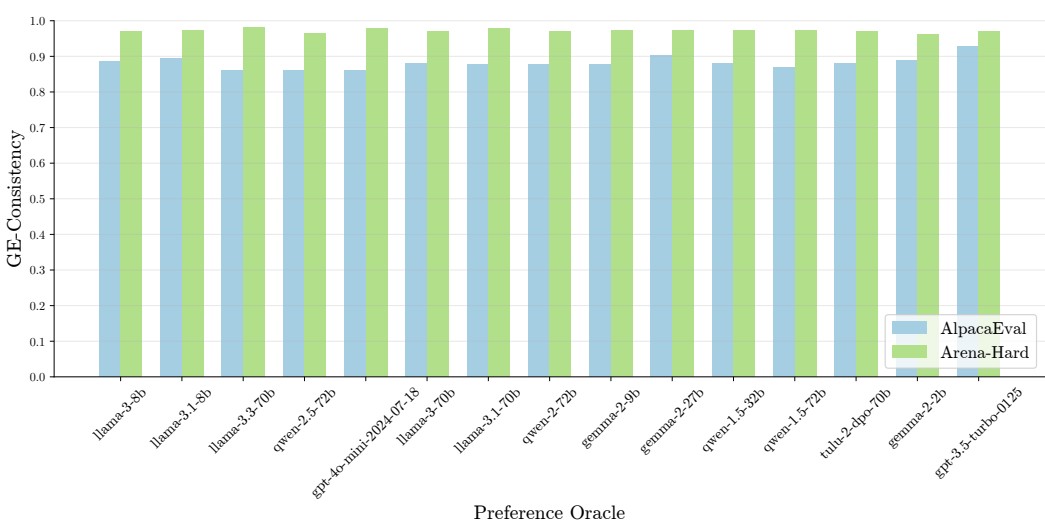

Figure 6: The stability of the GE-consistency with GPT-4o as the preference oracle when an arbitrary LLM is removed from the set of LLMs to be evaluated. Spearman's correlation between LLMs' generation and evaluation capabilities rankings is reported on the Y-axis. The X-axis shows the LLM that is removed from the LLM set to be evaluated.

