# OpenReview forum: "On Evaluating LLM Alignment by Evaluating LLMs as Judges"
_NeurIPS.cc/2025/Conference — NeurIPS 2025 poster_

### Official Review · Reviewer_fNcL · 2025-06-20

**Clarity:** 3
**Significance:** 3
**Originality:** 3
**Rating:** 4
**Confidence:** 4

**Summary:**

This paper investigates the Generation-Evaluation consistency (GE-consistency) of large language models (LLMs) and proposes a novel benchmarking paradigm, AlignEval, which evaluates LLM alignment by assessing their evaluation capabilities rather than directly judging their outputs. Experiments reveal a strong correlation between generation and evaluation rankings under specific conditions.

**Questions:**

- Will authors open source the proposed benchmark? This is very important.

**Ethical Concerns:**

["NO or VERY MINOR ethics concerns only"]

**Final Justification:**

Most of my concerns are addressed, and I believe my current rating reflects my assessment of this paper.

**Limitations:**

Yes

**Quality:**

3

**Strengths And Weaknesses:**

## Strength

This paper claims to be the first to conduct comprehensive study on GE consistency in LLMs, proposing a benchmark that bypasses direct output evaluation, offering a fresh perspective on alignment assessment. Their proposed benchmark, AlignEval, reduces evaluation costs (as no LLM-as-Judge needed) while aligning well with human preferences, making it cost-effective for real-world deployment. The paper is well-written, and I find it easy to follow and understand.

## Weakness

- Lack of human evaluation. I understand that reproducing the consistency analysis experiments in Section 3.2 by human annotators would be challenging and almost impossible. However, I believe it authors should at least try to validate the consistency hypothesis on a few LLMs using small-scale instructions. Since the entire paper fundamentally relies on the GE-Consistency assumption, I think additional supporting evidence would strengthen this core premise (this would be more persuasive though the current experiments across 15 LLMs are already relatively solid).

- Lack of more kinds of LLMs. The authors mainly study the System-1 model. Can the proposed benchmark be used for the System-2 model (Deepseek R1, O3, Qwen3 and so on)? This needs further verification.

- The author mentioned the importance of Consistency Filtering. However, this situation does not only occur with preference oracle. Did the author check how sensitive the tested LLM is to this situation? And how stable is the LLM's judgment output after multiple tests?

- A big concern lies in that the correlation on alpacaeval is significantly lower than arena-hard. Does this mean that the proposed benchmark cannot adequately validate the model's performance on open domain problems?

- This is related to the concern #4. Alignment is a big topic. Most of the author's experiments focus on the alignment of the preference of "helpfulness". Is there any similar experimental support for other preferences (safety and honesty)? I believe this is a very interesting and important question, which can make the assumptions and contributions of the article more solid.

---

> ### Author Rebuttal · Authors · 2025-07-31
>
> Thank you for your thoughtful review. We appreciate your recognition of the distinct perspective and contribution of our work.
>
> ---
> > “Lack of human evaluation. I understand that reproducing the consistency analysis experiments in Section 3.2 by human annotators would be challenging and almost impossible. However, I believe it authors should at least try to validate the consistency hypothesis on a few LLMs using small-scale instructions. Since the entire paper fundamentally relies on the GE-Consistency assumption, I think additional supporting evidence would strengthen this core premise (this would be more persuasive though the current experiments across 15 LLMs are already relatively solid).”
>
> We appreciate the suggestion. We agree that experiments with human annotations would be valuable, though they would require a significant amount of effort. We are grateful for your understanding of this difficulty. Even conducting such evaluations on a smaller set of instructions would be challenging, as we would still need to annotate outputs from multiple LLMs to compute rank correlations. Due to these constraints, we were unable to conduct such a study during the rebuttal period. However, we will aim to incorporate such analyses in the revised version, ideally using available human annotation datasets. If that turns out to be infeasible, we will include quantitative case studies with human evaluations to provide further intuition into GE-consistency.
>
> ---
> > “Lack of more kinds of LLMs. The authors mainly study the System-1 model. Can the proposed benchmark be used for the System-2 model (Deepseek R1, O3, Qwen3 and so on)? This needs further verification.”
>
> Thank you for the comment. We agree that extending evaluations to System-2/reasoning models is a valuable direction. However, several factors currently limit our ability to conduct a broader study on these models:
> (1) There are still relatively few reasoning-focused models available on the ChatBot Arena, and some checkpoints (e.g., earlier versions of Gemini-2.5-thinking) have become obsolete or inaccessible.
> (2) Reasoning models often require different evaluation configurations due to behavioral differences. For instance, generation temperature is either non-configurable or should not be set to zero.
> (3) Querying these models via API incurs substantial cost, especially when multiple generations are needed for stable estimates.
>
> Despite these limitations, we conducted a small-scale experiment using three reasoning models, o3-mini, o4-mini, and r1, as additional evaluated models in AlignEval with IFEval (AlignEval+). **The resulting correlations with ChatBot Arena rankings remained stable, suggesting that AlignEval can generalize to reasoning models.** Using the updated ChatBot Arena rankings that include these models, we obtained:
>
> |                          | AlignEval-GPT | AlignEval-Claude |
> |--------------------------|----------------|------------------|
> | w/o models | 0.932          | 0.940            |
> | w/ reasoning models    | 0.940          | 0.934            |
>
> We will expand on these results and include them in the revised manuscript.
>
> ---
> > “The author mentioned the importance of Consistency Filtering. However, this situation does not only occur with preference oracle. Did the author check how sensitive the tested LLM is to this situation? And how stable is the LLM's judgment output after multiple tests?”
>
> Thank you for the comment.
>
> 1. The self-inconsistency issue indeed occurs in models beyond the preference oracle. Below are the self-consistency rates (i.e., the proportion of consistent outputs across pairwise comparisons in two orders) for the 15 LLMs evaluated in Section 3.3. The results suggest that more capable LLMs tend to be more self-consistent.
>
> | Model                   | Score  |
> |-------------------------|--------|
> | qwen-2-72b              | 0.730  |
> | qwen-2.5-72b            | 0.720  |
> | gpt-4o-mini-2024-07-18  | 0.709  |
> | gemma-2-27b             | 0.642  |
> | gemma-2-9b              | 0.612  |
> | llama-3.1-70b           | 0.573  |
> | llama-3.3-70b           | 0.541  |
> | llama-3-70b             | 0.419  |
> | qwen-1.5-72b            | 0.493  |
> | qwen-1.5-32b            | 0.343  |
> | gpt-3.5-turbo-0125      | 0.307  |
> | tulu-2-dpo-70b          | 0.259  |
> | llama-3.1-8b            | 0.242  |
> | gemma-2-2b              | 0.225  |
> | llama-3-8b              | 0.093  |
>
> 2. We note that the self-inconsistency of the LLMs under evaluation does not compromise the reliability of AlignEval. Once examples are filtered based on the preference oracle’s self-consistency, any inconsistencies observed in the evaluated LLMs reflect their own limitations in producing reliable judgments.
>
> 3. Regarding the stability of the LLMs' judgment outputs across multiple runs: we set the temperature to 0 when evaluating LLMs as judges. Due to the limited time during the rebuttal period, we were unable to run repeated generations to assess output stability, but we plan to include such results in the revised version.
>
> Thank you for these helpful questions. We will update the manuscript accordingly.
>
> ---
> > “A big concern lies in that the correlation on alpacaeval is significantly lower than arena-hard. Does this mean that the proposed benchmark cannot adequately validate the model's performance on open domain problems?”
>
> 1. We would first like to clarify that GE-consistency is itself not a benchmark but a property we aim to study. As defined in Eq. 1, GE-consistency depends on both the choice of preference oracle and instruction set. In Section 3, we analyze how GE-consistency varies under different configurations of these two factors. Specifically, we use AlpacaEval and Arena-Hard as instruction sets due to their wide adoption. Our results show that the choice of instruction set indeed has a substantial impact on the level of GE-consistency.
>
> 2. Utilizing the above finding, we propose AlignEval, a benchmark whose goal is to align with human evaluations, using the LLM rankings from ChatBot Arena as the gold standard. AlignEval leverages the high GE-consistency observed on the Arena-Hard instruction set to enable efficient benchmarking. Since the ChatBot Arena rankings are aggregated from human votes on responses to user instructions, many of which are open-ended, the fact that AlignEval achieves a high level of correlation with ChatBot Arena suggests that it can effectively reflect human evaluations of alignment on open-ended tasks.
>
> 3.  **We have conducted further experiments** using the same setup as in Section 3.2 on WildBench [1], with GPT-4o as the preference oracle to evaluate 15 LLMs. Compared to Arena-Hard and AlpacaEval, WildBench offers a more balanced distribution of instruction types, for example, 14% involve creative writing, 12% involve reasoning, and 17% involve information seeking. On this more diverse instruction set, we observe a GE-consistency of 0.938 Spearman’s correlation, **suggesting that GE-consistency is a general pattern that holds across various types of instructions including open-domain tasks**.
>
> 4. We believe that the lower GE-consistency observed on AlpacaEval is partly due to the inherent noise in evaluating more open-ended and easier task instances. As evidence, Table 4 shows that the original AlpacaEval benchmark achieves only around 0.75 Spearman correlation with ChatBot Arena. This suggests that the AlpacaEval instruction set may have inherent limitations for robust LLM evaluation, and thus lower GE-consistency is expected.
>
> Thank you for the question. We will add a related discussion in the revised version to clarify this point.
>
> Reference
>
> [1] Lin, Bill Yuchen, et al. "WildBench: Benchmarking LLMs with Challenging Tasks from Real Users in the Wild." ICLR 2025
>
> ---
> > “... Most of the author's experiments focus on the alignment of the preference of "helpfulness". Is there any similar experimental support for other preferences (safety and honesty)? I believe this is a very interesting and important question, which can make the assumptions and contributions of the article more solid.”
>
> Thank you for the suggestion!
> 1. We would like to note that the preference oracle we used already takes into account various alignment aspects, such as safety and honesty. Specifically, the evaluation prompt (Figure 4 in the Appendix, Page 16) instructs: “You should prioritize evaluating whether the output honestly/precisely/closely executes the instruction, then consider its helpfulness, accuracy, level of detail, harmlessness, etc.”
> 2. We agree that more focused studies on safety or honesty would be interesting and valuable. However, incorporating them into the current submission is challenging, as it falls outside the scope of our present study and is limited by the lack of comprehensive human evaluation benchmarks (e.g., similar to Chatbot Arena) for these specific aspects, which we rely on as external ground truth. That said, we acknowledge the importance of these directions and will add related discussion in the revised manuscript.
>
> ---
> > “Will authors open source the proposed benchmark? This is very important.”
>
> Yes, we will open-source the proposed benchmark upon publication. We have already included the benchmark and the codebase for using it to evaluate models in the supplementary material. Thank you for the question. We will make this point clearer in the revised manuscript.

---

> > ### Comment · Area_Chair_Q6v6 · 2025-08-05
> >
> > Dear Reviewer fNcL,
> >
> > The authors have provided a response to your review. Could you please let us know if their response has addressed your concerns? This step is mandatory.
> >
> > Thank you, Area Chair

---

> > ### Comment · Reviewer_fNcL · 2025-08-05
> >
> > Thanks for your response. Most of my concerns are addressed, and I believe my current rating reflects my assessment of this paper.

---

### Official Review · Reviewer_5TaH · 2025-07-02

**Clarity:** 3
**Significance:** 2
**Originality:** 3
**Rating:** 4
**Confidence:** 3

**Summary:**

This paper explores whether LLMs exhibit consistent performance in two tasks: "generating aligned responses" and "evaluating whether responses are aligned." The authors propose GE-consistency, which uses Spearman correlation to measure the similarity of model performance across these two tasks. Evaluations on several mainstream LLMs lead to the conclusion that good generators are often also good evaluators. Based on this finding, the paper introduces the ALIGNEVAL evaluation method, which indirectly assesses a model’s generation ability by evaluating whether it can accurately judge human preferences, thereby effectively reducing evaluation costs. Finally, the method is compared with other evaluation approaches to validate its effectiveness.

**Questions:**

1. As shown in Figure 3, small models like Gemma-2B and LLaMA-3-8B do not perform well in terms of GE-consistency. Does this mean the method is only suitable for evaluating and ranking mid- to high-end models?
2. Since stronger models may perform better on all tasks, can the high correlation between generation and evaluation abilities be attributed to simultaneous linear improvement, rather than an intrinsic dependency?
3. You report that consistency on AlpacaEval is significantly lower than on Arena-Hard. Does this indicate that the ALIGNEVAL method is less suitable for open-domain tasks?

**Ethical Concerns:**

["NO or VERY MINOR ethics concerns only"]

**Final Justification:**

The authors have largely addressed my concerns on other points. However, their explanation regarding the distinction between correlation and causation, while partially reasonable, does not fully convince me. I have adjusted my score accordingly.

**Limitations:**

They discussed thelimitations of thier proposed benchmarking approach in section 5.

**Paper Formatting Concerns:**

No major formatting issues are identified.

**Quality:**

2

**Strengths And Weaknesses:**

Strengths:
1. The writing is good. The authors provides a novel evaluation perspective: by assessing whether an LLM can accurately judge, it indirectly verifies its generation ability, offering a new research direction for future work.
2. Based on the GE-consistency theory, the proposed ALIGNEVAL evaluation method can reduce computational and annotation costs.

Weakness:
1. In the GE-consistency experiments, only the Arena-Hard and AlpacaEval datasets were used. This seems insufficient, especially since the method performs well on the technical Arena-Hard dataset but poorly on the open-domain AlpacaEval dataset. More comprehensive experiments are needed to demonstrate generalizability.
2. If the proposed consistency theory holds, then ALIGNEVAL can indeed reduce evaluation costs for subsequent work. However, as the authors mention, it may be vulnerable to adversarial adaptation, where high scores do not necessarily indicate true capability. If this issue cannot be resolved, it will still be necessary to rely on evaluating the model’s output to fully verify its generalization ability, which may not actually reduce evaluation costs.
3. GE-consistency measures correlation, not causation. There is a potential confounding factor that "strong models perform well on all tasks," meaning that high scores across the board may obscure essential differences between the two abilities (uncertain).

---

> ### Author Rebuttal · Authors · 2025-07-31
>
> Thank you for your helpful comments and insightful questions.
>
> ---
> > “In the GE-consistency experiments, only the Arena-Hard and AlpacaEval datasets were used. This seems insufficient, especially since the method performs well on the technical Arena-Hard dataset but poorly on the open-domain AlpacaEval dataset. More comprehensive experiments are needed to demonstrate generalizability.” “You report that consistency on AlpacaEval is significantly lower than on Arena-Hard. Does this indicate that the ALIGNEVAL method is less suitable for open-domain tasks?”
>
> Thank you for the comment.
>
> 1. We would first like to clarify that GE-consistency is not a method but a property we aim to study. As defined in Eq. 1, GE-consistency depends on both the choice of preference oracle and instruction set. In Section 3, we analyze how GE-consistency varies under different configurations of these two factors. Specifically, we use AlpacaEval and Arena-Hard as instruction sets due to their wide adoption. Our results show that the choice of instruction set indeed has a substantial impact on the level of GE-consistency.
>
> 2. Utilizing the above finding, we propose AlignEval, a benchmark whose goal is to align with human evaluations, using the LLM rankings from ChatBot Arena as the gold standard. AlignEval leverages the high GE-consistency observed on the Arena-Hard instruction set to enable efficient benchmarking. Since the ChatBot Arena rankings are aggregated from human votes on responses to user instructions, many of which are open-ended, the fact that AlignEval achieves a high level of correlation with ChatBot Arena suggests that it can effectively reflect human evaluations of alignment on open-ended tasks.
>
> 3. Following your suggestion, **we conducted further experiments** using the same setup as in Section 3.2 on WildBench [1], with GPT-4o as the preference oracle to evaluate 15 LLMs. Compared to Arena-Hard and AlpacaEval, WildBench offers a more balanced distribution of instruction types, for example, 14% involve creative writing, 12% involve reasoning, and 17% involve information seeking. On this more diverse instruction set, we observe a GE-consistency of 0.938 Spearman’s correlation, **suggesting that GE-consistency is a general pattern that holds across various types of instructions including open-domain tasks**.
>
> 4. We believe that the lower GE-consistency observed on AlpacaEval is partly due to the inherent noise in evaluating more open-ended and easier task instances. As evidence, Table 4 shows that the original AlpacaEval benchmark achieves only around 0.75 Spearman correlation with ChatBot Arena. This suggests that the AlpacaEval instruction set may have inherent limitations for robust LLM evaluation, and thus lower GE-consistency is expected.
>
>
>
> We will update the revised manuscript with these results.
>
> Reference
>
> [1] Lin, Bill Yuchen, et al. "WildBench: Benchmarking LLMs with Challenging Tasks from Real Users in the Wild." ICLR 2025
>
>
> ---
> > “If the proposed consistency theory holds, then ALIGNEVAL can indeed reduce evaluation costs for subsequent work. However, as the authors mention, it may be vulnerable to adversarial adaptation, where high scores do not necessarily indicate true capability. If this issue cannot be resolved, it will still be necessary to rely on evaluating the model’s output to fully verify its generalization ability, which may not actually reduce evaluation costs.”
>
> Thank you for the comment. We agree that this is an important point, and we would like to expand our discussion in our conclusion section as follows, which we will update in the revised manuscript:
>
> 1. Since AlignEval is a proxy evaluation method, there is inevitably a trade-off between evaluation cost and accuracy/robustness. We believe the evaluation protocol we propose, combining AlignEval with IFEval, can help mitigate this risk, as it still requires the model to produce accurate outputs in response to instructions. In addition, targeted evaluations focusing on adversarial responses can be incorporated. These may be more cost-effective than the original LLM-judge-based evaluation while still addressing adversarial vulnerabilities.
>
> 2. AlignEval can be a valuable tool for benign model developers who do not have incentives to adversarially tune their models to exploit the metric. In such cases, AlignEval can serve as a cost-effective method for assessing model alignment during rapid development cycles.
>
> 3. To the best of our knowledge, this is the first work to formally introduce and study the notion of generation-evaluation consistency and its potential implications. Therefore, we consider the most important finding of this work to be that the evaluation capability of LLMs is itself an important evaluation aspect of LLMs’ alignment. We hope future work will build on this direction, further exploring topics such as the susceptibility of this evaluation paradigm to adversarial manipulation and the development of effective safeguards.
>
> ---
> > “GE-consistency measures correlation, not causation. There is a potential confounding factor that "strong models perform well on all tasks," meaning that high scores across the board may obscure essential differences between the two abilities (uncertain).” “Since stronger models may perform better on all tasks, can the high correlation between generation and evaluation abilities be attributed to simultaneous linear improvement, rather than an intrinsic dependency?”
>
> Thank you for this interesting and insightful comment. We conducted additional analyses to investigate this potential confounding factor and summarize below why it is not the sole reason for the high generation-evaluation consistency (GE-consistency) observed in our work:
>
> In Section 3.2.2, Figure 2, we report a Spearman’s correlation of 0.971 between LLMs’ generation and evaluation performance rankings (i.e., GE-consistency) on the instruction set of Arena-Hard using GPT-4o as the preference oracle. Under the same setting, the Spearman’s correlation between the generation performance ranking and the performance ranking on MixEval is 0.836, which is substantially lower than the observed GE-consistency. MixEval [1] evaluates alignment using a collection of tasks drawn from commonly used benchmarks such as MMLU and GSM8K, and therefore reflects a broader range of general capabilities. **The notably lower correlation suggests that the high GE-consistency we observe cannot be solely attributed to stronger models performing well across all tasks.**
>
> Thank you again for the thoughtful comment. We will incorporate these discussion points into the revised manuscript.
>
> Reference
>
> [1] Ni, Jinjie, et al. "Mixeval: Deriving wisdom of the crowd from llm benchmark mixtures." Advances in Neural Information Processing Systems 37 (2024): 98180-98212.
>
> ---
> > “As shown in Figure 3, small models like Gemma-2B and LLaMA-3-8B do not perform well in terms of GE-consistency. Does this mean the method is only suitable for evaluating and ranking mid- to high-end models?”
>
> Thank you for the question. We’d like to clarify that the evaluation in Figure 3 investigates GE-consistency when using preference oracles of varying capabilities. The results show that GE-consistency is lower when a less capable LLM is used as the oracle. This is intuitive: less capable LLMs may lack consistency in their evaluations and produce unreliable evaluation results. This does not mean that the method is only suitable for evaluating mid- to high-end models. Rather, it highlights that a reliable evaluation requires a sufficiently strong preference oracle. As shown in Section 3.2.2, when a strong LLM is used as the oracle, it can meaningfully assess and differentiate models across a broad range of capabilities, including weaker models, which tend to receive lower scores in both generation and evaluation performance.

---

> > ### Comment · Reviewer_5TaH · 2025-08-05
> >
> > The authors have largely addressed my concerns on other points. However, their explanation regarding the distinction between correlation and causation, while partially reasonable, does not fully convince me. I have adjusted my score accordingly.

---

### Official Review · Reviewer_FfNP · 2025-07-02

**Clarity:** 3
**Significance:** 3
**Originality:** 3
**Rating:** 4
**Confidence:** 4

**Summary:**

Current alignment evaluation practices predominantly focus on open-ended response tasks, typically relying on human annotations or leveraging stronger LLMs as preference oracles. However, certain LLMs themselves possess strong evaluation capabilities. This paper systematically investigates the relationship between an LLM’s generation and evaluation abilities within alignment tasks, and observe a notable level of generation-evaluation consistency (GE-consistency). Building upon this finding, this paper proposes AlignEval, a novel alignment benchmark that enables indirect evaluation of LLMs’ alignment with human preferences without assessing their generated outputs directly. AlignEval eliminates the need for human annotations or LLM judges, providing a fully automated and cost-effective solution for alignment evaluation.

**Questions:**

1. In the GE-consistency validation phase (Section 3), incorporating a few representative examples with manual evaluation would enhance the interpretability and persuasiveness of the observed consistency between generation and evaluation capabilities.
2. While the current GE-consistency findings are primarily based on rank correlation, which provides valuable global insights, it raises concerns about the reliability of the absolute scores reported in subsequent experiments.
3. In Figure 2, different models exhibit notable performance discrepancies across the two evaluation sets. It is recommended that the authors further investigate the underlying causes of this phenomenon, such as potential correlations with model scale or task generalization ability. A deeper analysis in this regard would strengthen the paper’s explanation of evaluation stability.

**Ethical Concerns:**

["NO or VERY MINOR ethics concerns only"]

**Final Justification:**

The author’s response did not alter my assessment of this work; therefore, my rating remains unchanged.

**Limitations:**

Yes

**Quality:**

3

**Strengths And Weaknesses:**

Strengths:
1. This paper addresses the consistency between the generation and evaluation capabilities of large language models (LLMs) by adopting a ranking-based consistency framework within alignment evaluation settings.
2. This paper proposes AlignEval, a novel benchmark that enables alignment evaluation without relying on human annotations or LLM-based judges, offering a fully automated, cost-effective, and efficient solution for assessing LLM alignment with human preferences.

Weaknesses:
1. The analysis of the paper’s main conclusions lacks sufficient depth, leaving certain critical questions unaddressed. This may undermine the perceived novelty and overall impact of the paper.

---

> ### Author Rebuttal · Authors · 2025-07-31
>
> We appreciate your thoughtful review and questions. We hope the following responses help clarify your concerns regarding the “certain critical questions unaddressed.”
>
> ---
> > “In the GE-consistency validation phase (Section 3), incorporating a few representative examples with manual evaluation would enhance the interpretability and persuasiveness of the observed consistency between generation and evaluation capabilities.”
>
> We agree that including representative examples with manual evaluation could enhance the interpretability of the observed GE-consistency. However, GE-consistency is defined as a rank correlation across multiple LLMs and diverse instructions, and a meaningful human evaluation would require substantial effort and time. Due to these constraints, we are unable to conduct a systematic investigation during the rebuttal period. That said, we will note this limitation and plan to provide such case studies in the revised manuscript.
>
> ---
> > “While the current GE-consistency findings are primarily based on rank correlation, which provides valuable global insights, it raises concerns about the reliability of the absolute scores reported in subsequent experiments.”
>
> Thank you for your comment. Our approach follows prior work (e.g., [1][2]) that constructs automatic LLM benchmarks using correlation with human evaluations as the primary metric, as it directly reflects the benchmark’s utility in ranking different LLMs. We agree that assessing the reliability of absolute score assignments could strengthen the evaluation. However, doing so would require analyzing the calibration of assigned scores, which involves specific methodologies beyond the current scope of our work. We will note this limitation in the revised manuscript.
>
> That said, as an initial step, we re-evaluated GE-consistency in Section 3.2.2 using Pearson’s correlation instead of Spearman’s, which considers the absolute values of the scores. The results still show a high level of GE-consistency: the Pearson correlation on AlpacaEval and Arena-Hard is 0.902 and 0.966, respectively, compared to the original Spearman correlations of 0.839 and 0.971.  **This suggests that GE-consistency still holds even when considering absolute scores.**
>
> References
>
> [1] Dubois, Yann, et al. "Length-controlled alpacaeval: A simple way to debias automatic evaluators." COLM 2024
>
> [2] Li, Tianle, et al. "From Crowdsourced Data to High-quality Benchmarks: Arena-Hard and Benchbuilder Pipeline." ICML 2025.
>
> ---
> > “In Figure 2, different models exhibit notable performance discrepancies across the two evaluation sets. It is recommended that the authors further investigate the underlying causes of this phenomenon, such as potential correlations with model scale or task generalization ability. A deeper analysis in this regard would strengthen the paper’s explanation of evaluation stability.”
>
> Thank you for this helpful suggestion. We believe the observed discrepancy is primarily due to differences in instruction difficulty: Arena-Hard is more challenging than AlpacaEval, as it contains more technical and complex instructions. This pattern is consistent with the official leaderboard performance of the two benchmarks. We will add a note regarding this in the revised manuscript.

---

> > ### Comment · Reviewer_FfNP · 2025-08-03
> >
> > Thank you for your response. However, it did not change my assessment of the paper, and I will maintain my original score.

---

### Official Review · Reviewer_VP2b · 2025-07-03

**Clarity:** 3
**Significance:** 3
**Originality:** 3
**Rating:** 4
**Confidence:** 4

**Summary:**

This paper investigates the relationship between large language models’ (LLMs) abilities to generate outputs aligned with human preferences and their abilities to evaluate such outputs. The authors formally define Generation–Evaluation Consistency (GE-consistency) as the Spearman correlation between LLM rankings by generation performance and by evaluation performance under a common preference oracle. They first measure GE-consistency using GPT‑4o on two benchmarks (AlpacaEval, Arena‑Hard), finding high correlations (ρ=0.839 and ρ=0.971) when filtering out inconsistent oracle judgments. They then study how GE-consistency varies with different oracles and demonstrate that stronger oracles yield higher consistency. Building on these findings, the paper proposes ALIGNEVAL, a benchmark that assesses LLM alignment by evaluating models in the judge role—using a fixed set of pairwise instances labeled by GPT‑4o or Claude-3.7-Sonnet—without requiring judges at test time. ALIGNEVAL matches or outperforms existing automatic benchmarks (e.g., AlpacaEval, Arena‑Hard) in correlating with ChatBot Arena human rankings (up to ρ=0.946), and when combined with IFEval, achieves ρ=0.94. The work highlights the importance of LLMs’ evaluation capabilities and introduces a scalable proxy for alignment assessment.

**Questions:**

1. Oracle Robustness: How does ALIGNEVAL perform if the preference labels are aggregated from multiple weaker oracles instead of a single strong one? Could ensembling or majority‑vote reduce bias and improve generality?

2. Broader Alignment Aspects: Can the GE‑consistency framework be extended to other evaluation tasks, such as toxicity or factuality judgments? What challenges arise when the evaluation criterion is less subjective or more specialized?

3. Theoretical Underpinning: Can the authors provide insight or hypotheses on why stronger evaluators tend to be stronger generators? For instance, is there evidence that underlying model capabilities (e.g., reasoning) drive both tasks?

**Ethical Concerns:**

["NO or VERY MINOR ethics concerns only"]

**Final Justification:**

The authors' response has cleared up my doubts. Since my score is already positive, I will not be changing it.

**Limitations:**

The authors acknowledge ALIGNEVAL is a proxy and vulnerable to adversarial fine‑tuning; combining with IFEval partially mitigates this. However, further discussion on multi‑oracle ensembling and detection of judge‑overfitting would strengthen the treatment of limitations.

**Paper Formatting Concerns:**

Figures and tables are clear, but captions could be more descriptive (e.g., explicitly stating which models are plotted).

**Quality:**

3

**Strengths And Weaknesses:**

Strengths
1. Significance & Originality: Introduces the novel concept of GE‑consistency, connecting two facets of LLM behavior—generation and evaluation—and demonstrates its high correlation under strong oracles (Section 3)

2. Benchmark Innovation: Proposes ALIGNEVAL, which repurposes judge instances to evaluate alignment without on-the-fly judging, reducing computational cost and preserving high correlation with human preferences (Section 4)

3. Comprehensive Empirical Study: Evaluates 15–23 diverse LLMs across multiple oracles and instruction sets, with clear ablations on filtering and oracle strength (Figures 2, Table 1)

4. Practical Impact: Demonstrates that ALIGNEVAL combined with IFEval matches or exceeds performance of established benchmarks, offering a cost‑effective alternative for both researchers and practitioners.

Weaknesses
1. Dependence on Oracle Quality: High GE‑consistency relies on very strong oracles (e.g., GPT‑4o); weaker oracles yield substantially lower consistency, limiting generalizability without access to frontier models (Figure 3)

2. Oracle Self‑Bias: ALIGNEVAL exhibits self‑preference bias (models rank their own version highly), which may inflate correlations unless multiple oracles are used (Section 4.3)

3. Limited Theoretical Insight: While empirical correlations are strong, the paper offers limited theoretical explanation for why GE‑consistency holds or how it evolves during training.

---

> ### Author Rebuttal · Authors · 2025-07-31
>
> Thank you for your helpful comments. We are encouraged by your recognition of the impact and novelty of our work.
>
> ---
> > “Dependence on Oracle Quality: High GE‑consistency relies on very strong oracles (e.g., GPT‑4o); weaker oracles yield substantially lower consistency, limiting generalizability without access to frontier models (Figure 3)”
>
> Thank you for the comment. Our investigation indeed shows that achieving high GE-consistency requires strong preference oracles. However, **due to the design of our proposed benchmark, AlignEval, this reliance on frontier models does not pose a bottleneck for generalizability.** Specifically, AlignEval leverages existing preference annotations from strong oracles to evaluate LLMs as evaluators, so **constant access to frontier models is not required once the benchmark is constructed.** This also enables the reuse of expensive human annotations using the same methodology. We will expand on this point in the revised manuscript.
>
> ---
> > “Oracle Self‑Bias: ALIGNEVAL exhibits self‑preference bias (models rank their own version highly), which may inflate correlations unless multiple oracles are used (Section 4.3)”
>
> We note that this self-bias does not necessarily inflate the correlations. In cases where earlier versions of a model are rated poorly by human evaluators, the model’s self-preference may actually lower the correlation. Thus, rather than introducing a systematic bias, the self-bias is more likely to increase the variance of the correlations. Furthermore, as discussed in Lines 318-322, AlignEval is still able to rank gemini-2.0-flash, the top system according to human evaluation, among the top two, despite the presence of potential self-bias.
>
> ---
> > “Limited Theoretical Insight: While empirical correlations are strong, the paper offers limited theoretical explanation for why GE‑consistency holds or how it evolves during training.” “Theoretical Underpinning: Can the authors provide insight or hypotheses on why stronger evaluators tend to be stronger generators? For instance, is there evidence that underlying model capabilities (e.g., reasoning) drive both tasks?”
>
> These are indeed interesting questions. This work is motivated by the hypothesis that an LLM capable of generating outputs well aligned with human preferences should also be better at evaluating whether an output aligns with those preferences. Our investigation confirms that such a property, generation-evaluation (GE) consistency, emerges when using a strong preference oracle. To the best of our knowledge, this is the first work to formally introduce and examine GE-consistency and its potential implications. We hope future work will build on this direction, for instance, by exploring the insightful questions you have raised regarding its theoretical foundations and training dynamics.
>
> ---
> > “Broader Alignment Aspects: Can the GE‑consistency framework be extended to other evaluation tasks, such as toxicity or factuality judgments? What challenges arise when the evaluation criterion is less subjective or more specialized?”
>
> Thank you for these thoughtful questions.
> 1. We would like to note that the preference oracle used in our study already considers multiple alignment aspects, such as toxicity and factuality. Specifically, the evaluation prompt (Figure 4 in the Appendix, Page 16) instructs the model to “prioritize evaluating whether the output honestly/precisely/closely executes the instruction, then consider its helpfulness, accuracy, level of detail, harmlessness, etc.”
> 2. We agree that more targeted studies focusing specifically on aspects like toxicity or factuality are valuable. However, incorporating such studies is challenging within the scope of the current work, due to both the breadth of our focus and the lack of comprehensive human evaluation benchmarks (e.g., similar to Chatbot Arena) dedicated to these specific dimensions. That said, we recognize the importance of these directions and will include a discussion of them in the revised manuscript.
>
> ---
> > “Oracle Robustness: How does ALIGNEVAL perform if the preference labels are aggregated from multiple weaker oracles instead of a single strong one? Could ensembling or majority‑vote reduce bias and improve generality?”
>
> Thank you for these thoughtful questions. To explore this, we conducted a pilot study where we aggregated preference labels from two weaker oracles, Qwen3-8B and Qwen3-14B, using the AlignEval+ framework (i.e., with IFEval). The results show that this ensemble approach improves correlation with human judgments: used individually, Qwen3-8B and Qwen3-14B achieve Spearman’s correlations of 0.853 and 0.865 with Chatbot Arena, respectively. When combined via majority vote, the ensemble reaches a higher correlation of 0.876. Moreover, we believe that ensembling can help reduce potential self-bias introduced by individual oracles, as discussed above. We will report this result and expand on it in the revised manuscript.

---

> > ### Comment · Reviewer_VP2b · 2025-08-04
> >
> > Thank you for your reply. Since my score is already positive, I will not be changing it.

---

### Note · Authors · 2025-08-16

We thank all the reviewers for their valuable reviews and constructive comments, and the Area Chair for facilitating the author-reviewer discussion and overseeing the review process. We appreciate the reviewers’ recognition of the strengths of our submission. Specifically, all the reviewers have noted the novelty and significance of our studied topic, the generation-evaluation consistency among LLMs, which was described as “a novel concept connecting two facets of LLM behavior” (Reviewer VP2b), “a novel evaluation perspective” (Reviewer 5TaH), and “a fresh perspective on alignment assessment” (Reviewer fNcL). The reviewers also agree on the practical value and the novelty of our proposed benchmark, AlignEval, highlighting that it is cost-effective as it does not require LLMs-as-Judges thanks to the benchmark design inspired by our findings regarding generation-evaluation consistency, which offers "a fully automated, cost-effective, and efficient solution for assessing LLM alignment with human preferences" (Reviewer FfNP).

We are grateful for the reviewers’ replies to our rebuttal, and we are especially encouraged by the feedback from Reviewer 5TaH and Reviewer fNcL indicating that their concerns have been largely addressed. We will carefully incorporate the reviewers’ valuable comments and suggestions into our revised manuscript. In particular, we plan to add the additional evaluation results on WildBench which have been presented during the discussion period. These results further demonstrate that the generation-evaluation consistency studied in our work is a general pattern that holds across various types of instructions including open-domain tasks, which has helped address the concern of the reviewers regarding the lower generation-evaluation consistency observed on AlpacaEval compared to Arena-Hard.

---

### Decision · Program_Chairs · 2025-09-17

**Decision:**

Accept (poster)

**Comment:**

# Summary
This paper explores whether LLMs exhibit consistent performance in two tasks: "generating aligned responses" and "evaluating whether responses are aligned." The authors propose GE-consistency, which uses Spearman correlation to measure the similarity of model performance across these two tasks. Evaluations on several mainstream LLMs lead to the conclusion that good generators are often also good evaluators. Based on this finding, the paper introduces the ALIGNEVAL evaluation method, which indirectly assesses a model’s generation ability by evaluating whether it can accurately judge human preferences, thereby effectively reducing evaluation costs. Finally, the method is compared with other evaluation approaches to validate its effectiveness.

# Strengths
* Significance & Originality: Introduces the novel concept of GE‑consistency, connecting two facets of LLM behavior—generation and evaluation—and demonstrates its high correlation under strong oracles
* Benchmark Innovation: Proposes ALIGNEVAL, which repurposes judge instances to evaluate alignment without on-the-fly judging, reducing computational cost and preserving high correlation with human preferences
* Comprehensive Empirical Study: Evaluates 15–23 diverse LLMs across multiple oracles and instruction sets, with clear ablations on filtering and oracle strength

# Weaknesses
* Dependence on Oracle Quality: High GE‑consistency relies on very strong oracles (e.g., GPT‑4o); weaker oracles yield substantially lower consistency, limiting generalizability without access to frontier models
* Oracle Self‑Bias: ALIGNEVAL exhibits self‑preference bias (models rank their own version highly), which may inflate correlations unless multiple oracles are used
* Limited Theoretical Insight: While empirical correlations are strong, the paper offers limited theoretical explanation for why GE‑consistency holds or how it evolves during training.

# Concerns addressed during Author Response Period
* Correlation on alpacaeval is significantly lower than arena-hard. Does this mean that the proposed benchmark cannot adequately validate the model's performance on open domain problems?
* The authors run additional experiments which suggest that GE-consistency is a general pattern that holds across various types of instructions including open-domain tasks.
The preference oracle in the paper already takes into account various alignment aspects, such as safety and honesty.

# Overall
The paper proposes ALIGNEVAL, reducing the computational cost of evaluating the alignment capabilities of LLMs. The approach is novel and interesting. There are a few remaining concerns as the GE-consistency relies on a strong oracle quality and self-biases.